# Exploring how infectious diseases control interventions are delivered in low-income urban communities in low- and middle-income countries: A scoping review protocol

Hammed O. Mogaji[1], Ifrah A. Fatah[2], Scott C. Dorris[3], Margaret C. Baker[4]*

1 Department of Behavioral and Applied Sciences, Marian University, Indianapolis, Indiana, United States of America, 2 Georgetown University, Washington, DC, United States of America, 3 Dahlgren Memorial Library, Georgetown University, Washington, DC, United States of America, 4 Department of Global Health, Georgetown University, Washington, DC, United States of America

* mcb93@georgetown.edu

## Abstract

Infectious diseases thrive in densely populated low-income urban communities where living conditions, hygiene, and access to health services are sub-optimal. Information on how to adapt infectious diseases control interventions to be effective in these settings is limited. The proposed scoping review will follow Joanna Briggs Institute (JBI) guidelines and the PRISMA methodology adapted for scoping reviews (PRISMA-ScR). Research questions were created using the Person, Concept, Context (PCC) framework. In this study, person refers to low-income urban communities, concept refers to community-based infectious disease interventions adapted to their setting, and the context is low- and middle-income countries. Relevant papers published after 2000 will be retrieved using a search strategy that incorporates all the PCC topics. Searches will be made in four databases: MEDLINE, Embase, Global Health (all via Ovid), and Web of Science. Duplicates will be identified and removed with EndNote 20, and all articles will be uploaded into Covidence for screening in two stages using pre-defined inclusion and exclusion criteria. Quantitative and qualitative data will be extracted into Excel spreadsheets and analyzed. Our review will identify different approaches that have been taken to adapt infectious disease interventions in low-income urban communities of LMICs that can be further tested in future programs and studies. We will also highlight methods that have been used to study such interventions in their context.

## Introduction

Urbanization has accelerated rapidly in this century, with approximately 57% of the world's population now living in cities [1]. Furthermore, the global urban population is

**Data availability statement:** No datasets were generated or analysed during the current study. All relevant data from this study will be made available upon study completion.

**Funding:** The author(s) received no specific funding for this work.

**Competing interests:** The authors have declared that no competing interests exist.

projected to reach 6.25 billion by 2050 [2]. This growth is driven by multiple factors, including government policies on public housing, land use, property ownership, and economic opportunities [3,4]. To date, the largest expansion of urban populations has been focused on low and middle-income countries (LMICs) [5]. This rapid pace and scale of urbanization has resulted in significant pressure on available infrastructure and resources, with expansion into areas previously considered uninhabitable, often referred to as informal settlements or slums and are characterized by inadequate housing conditions, poor health and safety, and fewer work opportunities [4,6,7].

The impact of urbanization on the spread of infectious diseases is highly contextualized. On one hand, urbanization can reduce mortality and morbidity caused by infectious agents in areas where living conditions, hygiene, and access to health services are optimal [8]. On the other hand, rapid urbanization has also resulted in the growth of informal settlements [9], where infectious diseases thrive. Factors associated with the spread of infectious diseases in urban areas, especially low-income urban communities, have been summarized by Alirol *et al* [8] and Neiderud *et al* [10]. High population density allows air-borne infections like influenza, measles, and *Mycobacterium tuberculosis* to rapidly spread [11]. Inadequate water and sanitation services contribute to diarrhoeal diseases caused by cholera and *E. coli* [12,13], to the transmission of soil-transmitted helminthiasis and other intestinal parasites [14], and to the proliferation of rodents such as rats which can carry *Leptospira interrogans* [15]. Accumulated waste can provide breeding sites for vectors like sand flies that carry parasites causing leishmaniasis and of *Aedes* mosquitoes that are vectors for dengue, yellow fever, and chikungunya [16].

While many interventions have proven to be effective in preventing infectious diseases [17], implementation failure, including in urban settings, has also been documented. For example, immunization programs have reported challenges in reaching urban dwellers due to working with large populations that are transient in nature – with inconsistent employment patterns, and multicultural beliefs and practices [18]. Similarly, mass drug administrations (MDA) for controlling neglected tropical diseases (NTDs) in Africa and the Americas, have had lower reported coverage in urban settings due to the heterogeneity that exists within urban environments [19,20]. The success of immunization, contraception, and antenatal care services was also reported to be sub-optimal in urban slum areas of Chandigarh in the North of India. [18,21]. Community health interventions in urban settings, especially, in low-income communities, must therefore contend with population mobility, informal housing, and cultural diversity, which together add complexity to the delivery of services.

These persistent challenges highlight the need to understand how interventions are being adapted in practice. Yet, despite this recognition, there is limited synthesized evidence on adaptations to low-income urban communities in LMICs. A recent review begun to address related questions by mapping the types of community-based interventions delivered in informal urban settlements, however they stopped short of providing insight on how interventions were adapted to this setting [17]. These gaps demonstrate the need for a review that specifically identifies approaches taken to adapting interventions and how these were studied.

This review will address that gap – across multiple infectious diseases – by concentrating on a subset of population-based mass interventions characterized by including one-on-one interactions between a distributor and a recipient within the community. This delivery mode, often referred to as 'campaigns', is commonly used for vaccinations, mass drug administration (MDA), screening and test-and-treat initiatives, and distribution of products such as bed nets, condoms, and facemasks. This focus aligns with the methodological approach taken in a previous scoping review focused on a different marginalized population, namely ethnically, racially, and religiously marginalized populations. [22]

## Aim and objectives of the review

The aim of this scoping review is to map and synthesize evidence on how infectious disease interventions have been adapted for delivery in low-income urban communities in LMICs [23]. In line with this aim, our objectives are to: (1) catalogue the types of adaptations reported; (2) examine how these adaptations relate to context; (3) assess the extent and distribution of the available evidence; and (4) review the study designs and methods used to investigate these adaptations.

## Materials and methods

### Protocol design

Scoping reviews use standardized methods, taking a systematic approach, and are the methodology of choice when the aim is to get a broad overview of a potentially diverse body of literature [24]. This protocol documents how we will conduct a scoping review using the Joanna Briggs Institute (JBI) guidelines [24], and the Preferred Reporting Items for Systematic Review and Meta-Analysis extension for Scoping Reviews (PRISMA-ScR) guidelines [25]. The PRISMA ScR checklist, completed for this study, has been included as Supplementary File 1.

In this review we used the Person, Concept, Context (PCC) framework to guide the formation of research questions and search strategies [26]. Our population will include those living in low-income urban communities. The concept will be adaptations of infectious disease interventions characterized by one-on-one interactions between a distributor and a recipient within the community. The context will be low-middle-income countries as defined by the World Bank [23].

### Eligibility criteria

Our inclusion and exclusion criteria is detailed in Table 1. We consider studies published from 2000 relevant because many large-scale infectious disease control initiatives in LMICs - including global immunization campaigns, scale-up of HIV/TB programs, and mass drug administration for NTDs - expanded significantly after 2000, often with increased attention to delivery in urban contexts. We exclude studies based in rural areas, not because these studies are less important, but to align with our stated objective.

### Information sources and search strategy

Relevant articles published from the year 2000 onwards until the date of search will be retrieved from MEDLINE, Embase, Global Health (all via Ovid), and Web of Science with the goal to identify articles from both medical and social sciences fields. The search strategy incorporates both database-specific subject headings and keywords that represent all three PCC topics: low-income urban communities, infectious disease interventions, and LMICs. As there is no universally accepted global definition of urban, we will use search terms like "urban," "peri-urban," "city," "municipality," "urbanization," and "township." used in other literature reviews on this subject. [17,27,28]. Terms related to low-income urban communities including slums, informal settlements, street dwellers, are also used to select for the population of interest. As our concept focuses on mass infectious disease interventions that require interpersonal interaction between those delivering a health product and those receiving it, we included search terms related to vaccine, mass drug administration, and mass

**Table 1. Inclusion and exclusion criteria.**

| Framework | Items | Inclusion | Exclusion |
|---|---|---|---|
| Population | Communities | Studies concerning low-income urban communities. | High income urban communities and rural communities. |
| Concept | Type of intervention | Studies on interventions that require interpersonal contact in a community setting, specifically: immunizations, mass drug administration, deworming, screening, contact tracing, and distribution of commodities (e.g., condoms, facemasks, bed nets). | Studies on interventions that do not rely on one-one interaction. This includes interventions reaching many people simultaneously with no interpersonal interaction in the delivery (e.g., mass communication and health education campaigns) and interventions that rely on environmental modifications (e.g., spraying for vector control or water and sanitation projects). Interventions delivered exclusively in facility settings. |
| Concept | Information on adaptation | Studies that describe how the intervention was adapted to low-income urban communities, even if very limited. | Studies that did not describe how the intervention was adapted to low-income urban communities. |
| Concept | Reported outcome | Studies must report at least one outcome (e.g., implementation, clinical/epidemiologic, economic, or experiential) indicating that the intervention was delivered. | Studies that did not report any implementation or intervention outcome. |
| Context | Countries | Studies implemented in Low- and Middle-Income Countries (LMICs). | Studies implemented in High Income Countries (HICs). |
| Context | Study design and type of publication | Research studies that are peer-reviewed and published. Can be qualitative, quantitative or mixed methods, descriptive or analytic, and primary or secondary research. | Literature reviews, textbooks, dissertations, editorials, opinion pieces, and grey literature. |
| Context | Publication language | Studies reported in all languages. | |

screening or testing for infectious diseases. The search strategy for Ovid MEDLINE is shown in [see S2 File]. The search strategy was developed by a university librarian specialist and peer-reviewed with the other authors, then tested to obtain an optimal balance between allowing for a broad scope and feasibility. These search terms were subsequently translated for the remaining databases.

## Study selection

All retrieved articles will be uploaded into Covidence Software for screening. Duplicates will be identified and removed with EndNote 20 (Clarivate Analytics, Philadelphia, PA). Article screening will be performed by four trained research assistants. In the first phase, titles and abstracts of at least 200 articles will each be independently reviewed by two researchers – until an inter-rater reliability of 95% agreement is achieved. Discrepancies will be resolved through discussion. Once this threshold is reached, screening will proceed with one researcher per article. Articles included in the title and abstract stage of screening will then undergo full-text review, with the same eligibility criteria used to determine relevance. A PRISMA flow diagram will be created to summarize the results of the search and reasons for exclusion. Reviewers will continue to meet every 1–2 weeks to discuss the studies, including any articles for which there is uncertainty on whether to include it.

## Data extraction

The data extraction process will be conducted by two researchers who will extract data from assigned articles. This review will be checked by a second researcher. A standardized data extraction form in Microsoft Excel will be used to capture both quantitative and qualitative information. The extraction form will include information on the topics presented in Box 1 as well as bibliographic information (including author, year of publication, and language) and information on methods and outcomes. The extraction framework was developed with reference to the Expert Recommendations for Implementing Change (ERIC) framework [29] and the Practical, Robust Implementation and Sustainability Model (PRISM) [30], both of which have been widely applied in implementation research.

**Box 1: Information to be extracted about the interventions and context**

**Intervention:**

• What the intervention is, e.g., bed net distribution, health education campaign, screening

• What the intervention type is, e.g., preventive, diagnostic, curative

• Why the intervention was needed

• Intervention goal, including disease(s) targeted

**Context:**

• Where implemented: Countries, sub-country, number of sites

• Community setting environment: socio- economic, geography, political

• Population receiving intervention

**Implementation and adaptation made to the context:**

• Delivery modalities with explanation on why selected including point of delivery and timing

• How the community was initially approached and how they were engaged during each stage of the study

• What organization(s) led delivery of the intervention and the roles of staff within these organizations

• Other stakeholders and their role

• Processes put in place to coordinate between all stakeholders

• Adoption of intervention (e.g., number of workshops, amount of product delivered) and fidelity to the original plan

• How decisions were made related to the context

• Facilitators: what worked well and why

• Barriers: what did not work, why, and how these were overcome

• Sustainability: how long were the interventions delivered

• Costs and funding

**Analysis of results**

We will analyze data using both descriptive summaries and thematic narrative analysis. For descriptive summaries we will present tables summarizing basic study characteristics: country (and region), year of publication, intervention type, type of urban setting (e.g., informal settlement vs other), and population demographics. All elements of mixed or multi-component studies will be extracted. We will not attempt meta-analysis or effectiveness estimates as this review focuses on identifying how interventions have been adapted and how these adaptations have been studied.

Thematic analysis will focus on how interventions were adapted; what was reported to work and not work; barriers and facilitators; how stakeholders were engaged; context-driven decisions; and the socio-economic, geographic, and political variables that shaped adaptation. Analysis will be qualitative and conducted by the two senior researchers. All textual data

will first be carefully reviewed and emerging themes identified. Additional columns will then be created in the extraction spreadsheet, and thematic labels (codes) manually entered. Results will be synthesized within each theme, with a detailed description of common trends as well as notable exceptions. To complement researcher perspectives, ChatGPT will be queried to identify potential additional themes and to further explore the data within each theme. Any information provided by ChatGPT will be manually cross-checked against the original data sources before inclusion.

### Status and timeline of the study

The scoping review will be conducted over a six-month period, from August 2025 to January 2026. Development of the search strategy will take place in August, followed by the initial screening of titles and abstracts within the same month. Full-text screening will occur in September, followed by data extraction in October. Data analysis and interpretation will take place in November.

### Protocol deviation

Any deviations from the described protocol will be documented and transparently reported in future publications and presentations of study results. The research team will ensure that any deviations are justified, minimized, and do not compromise the integrity or objectives of the study.

### Data-sharing

In line with open science principles, we will make the final data extraction sheet openly available through a public repository such as the Open Science Framework.

### Ethical considerations

Ethical approval was not required because the study does not include participants. However, this study is in accordance with the Declaration of Helsinki [31] and Ethical Guidelines for Nursing Research in the Nordic Countries [32].

### Patient and public involvement

None

## Discussion

Our overarching aim in this review is to map and synthesize evidence on how community-based infectious disease interventions have been adapted for delivery in low-income urban communities in LMICs. The findings will enhance understanding of how such interventions might be adapted for these communities and how adaptations relate to context. The review will highlight key facilitators and barriers to implementing adaptations as well as methods used to study these – providing valuable insights to guide future research and practice in similar contexts.

The strengths of this review include its methodological rigor, with procedures designed to minimize selection bias. These are reinforced by the authors' prior experience conducting a related review among other marginalized groups [22], collaboration with a senior trained librarian, and the inclusion of articles published in multiple languages, supported by the team's diverse linguistic capabilities. We also employed a pilot-screening approach to strengthen selection procedures, drew upon the team's extensive experience delivering infectious disease interventions in LMICs across three world regions, and applied established frameworks for data extraction and analysis.

A potential limitation of our review is the availability of studies that meet our inclusion criteria. The delivery of infectious disease interventions is usually led by health program implementers and frontline workers, with relatively few academic researchers engaged in sustained collaboration with these groups. More broadly, implementation research – and the

networks linking academics with program practitioners – remains a relatively nascent field. As a result, the pool of eligible studies may be limited; however, documenting this gap will itself be an important contribution.

## Dissemination plan

The results from the scoping review will be submitted for publication in a peer-reviewed journal also be prepared for submission to conferences.

## Conclusion

Our review will provide critical insights into approaches used to adapt infectious disease interventions for marginalized populations in LMIC low-income urban communities. By identifying effective strategies and highlighting contextual facilitators and barriers, the study will inform both future research and innovative program design. Ultimately, this work will contribute to strengthening the evidence base for sustainable, culturally appropriate, and impactful infectious disease interventions in resource-constrained urban settings.

## Supporting information

**S1 File. PRISMA ScR Checklist.**
(DOCX)

**S2 File. Search terminology.**
(DOCX)

## Author contributions

**Conceptualization:** Margaret C. Baker.

**Data curation:** Hammed O. Mogaji, Ifrah A. Fatah, Scott C. Dorris, Margaret C. Baker.

**Formal analysis:** Hammed O. Mogaji, Margaret C. Baker.

**Investigation:** Hammed O. Mogaji, Ifrah A. Fatah, Scott C. Dorris, Margaret C. Baker.

**Methodology:** Hammed O. Mogaji, Ifrah A. Fatah, Scott C. Dorris, Margaret C. Baker.

**Project administration:** Hammed O. Mogaji, Scott C. Dorris, Margaret C. Baker.

**Resources:** Hammed O. Mogaji, Scott C. Dorris, Margaret C. Baker.

**Software:** Hammed O. Mogaji, Scott C. Dorris.

**Supervision:** Hammed O. Mogaji, Margaret C. Baker.

**Validation:** Hammed O. Mogaji, Ifrah A. Fatah, Margaret C. Baker.

**Visualization:** Hammed O. Mogaji, Margaret C. Baker.

**Writing – original draft:** Hammed O. Mogaji, Ifrah A. Fatah, Scott C. Dorris, Margaret C. Baker.

**Writing – review & editing:** Hammed O. Mogaji, Ifrah A. Fatah, Scott C. Dorris, Margaret C. Baker.

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
