## [Decision Letter · Decision Letter 0]

22 Aug 2025

Dear Dr. Mogaji,

Thank you for submitting your manuscript to PLOS ONE. After careful consideration, we feel that it has merit but does not fully meet PLOS ONE’s publication criteria as it currently stands. Therefore, we invite you to submit a revised version of the manuscript that addresses the points raised during the review process.

We look forward to receiving your revised manuscript.

Kind regards,

Mickael Essouma, M. D.

Academic Editor

PLOS ONE

Journal Requirements:

4. Please amend your authorship list in your manuscript file to include author Hammed Oladeji Mogaji

5. Please amend the manuscript submission data (via Edit Submission) to include author Mogaji, H.O

Additional Editor Comments :

In addition to the reviewer's comments, I advise the authors to thoroughly revise their manuscript showing creativity so that the aim of their upcoming scoping review and the methodology used to debelop it differ substantially from two recent major reviews in the same research domain:

1. Epling et al. Lancet Infect Dis. 2025 May;25(5):e269-e279. doi: 10.1016/S1473-3099(24)00744-8.

2. Shafique et l. Syst Rev. 2024 Oct 4;13(1):253. doi: 10.1186/s13643-024-02651-9.

More details are available in the word document PONE-D-25-40772_reviewed by Mickael Essouma attached to this decision letter.

Mickael Essouma, M.D.

Reviewers' comments:

Reviewer's Responses to Questions

**Comments to the Author**

1. Does the manuscript provide a valid rationale for the proposed study, with clearly identified and justified research questions?

Reviewer #1: Yes

2. Is the protocol technically sound and planned in a manner that will lead to a meaningful outcome and allow testing the stated hypotheses?

Reviewer #1: Yes

3. Is the methodology feasible and described in sufficient detail to allow the work to be replicable?

Reviewer #1: Yes

4. Have the authors described where all data underlying the findings will be made available when the study is complete?

Reviewer #1: No

5. Is the manuscript presented in an intelligible fashion and written in standard English?

Reviewer #1: Yes

You may also provide optional suggestions and comments to authors that they might find helpful in planning their study.

Reviewer #1: I have attached a word document with my comments. Authors must revise their manuscript and the manuscript can be accepted.

**Do you want your identity to be public for this peer review?** For information about this choice, including consent withdrawal, please see our Privacy Policy

Reviewer #1: **Yes: ** Tracy Zhandire

---

## [Author Response · Author response to Decision Letter 1]

31 Oct 2025

Additional Editor Comments :

C1: In addition to the reviewer's comments, I advise the authors to thoroughly revise their manuscript showing creativity so that the aim of their upcoming scoping review and the methodology used to debelop it differ substantially from two recent major reviews in the same research domain:

1. Epling et al. Lancet Infect Dis. 2025 May;25(5):e269-e279. doi: 10.1016/S1473-3099(24)00744-8.

2. Shafique et l. Syst Rev. 2024 Oct 4;13(1):253. doi: 10.1186/s13643-024-02651-9.

More details are available in the word document PONE-D-25-40772_reviewed by Mickael Essouma attached to this decision letter.

Response to Reviewer

We appreciate the reviewer’s suggestion to clarify how our scoping review differs from recent reviews in this area, particularly Epling et al. (Lancet Infect Dis, 2025) – authored by M Baker, also an author of this review - and Shafique et al. (Syst Rev, 2024).

This study focuses on populations living in low income urban settings in LMICs while the Epling et al. study focused on populations marginalized by ethnicity, race, or religion - globally. These are intended to be part of a series of papers exploring adaptations of interventions for marginalized populations – with each paper focusing on a different marginalized population, while looking across diseases.

Shafique et al. synthesized community-based interventions for infectious disease control in urban informal settlements in LMICs, identifying a range of interventions. However, their review did not examine how these interventions were adapted to fit urban poor settings. Our study addresses this gap by focusing specifically on adaptation processes and mechanisms of delivery in urban low income settings in LMICs.

This paper is therefore distinct in two ways:

(1) it hones in on population living in low income urban settings, in LMICs—a group both underserved in research and highly relevant to current global health priorities;

(2) it explicitly examines adaptation of interventions to these settings, rather than simply listing interventions; and

It therefore builds on but does not duplicate the earlier reviews, contributing to a planned series of complementary papers that map how delivery strategies vary across different marginalized populations and contexts.

We will update both the Introduction and Discussion to make these distinctions clearer, citing the two prior reviews directly and specifying how this study extends, rather than duplicates, their contributions.

C2: Editors comments on Title within the text

● Do you wish to focus on ID outbreaks or on all types of IDs? This question stems from the fact that IDs are ubiquitous whereas the concern of IDs with regard to rapid urbanization in LMICs is most often raised in the context of ID outbreaks because ID outbreaks are more likely to occur in urban localities of so-called LMICs than in urban localities of purported developed countries.

Focusing on ID outbreaks could also help the authors make a review differing from this 2025 scoping review (PMID: 39922209, DOI: 10.1016/S1473-3099(24)00744-8) and from this 2024 systematic review (SR; https://doi.org/10.1186/s13643-024-02651-9) especially since a SR is seen as a more useful form of evidence synthesis than a scoping review and the lag time required to update that SR may not have been reached yet (https://doi.org/10.1136/bmj.i3507). Also see https://doi.org/10.1038/s41579-021-00639-z. If you focus on ID outbreaks, then this should be specified throughout the manuscript (from the title to the end of the manuscript), not on IDs in general.

We thank the editor for this suggestion. To clarify, our review is not restricted to outbreaks but instead examines all infectious disease interventions delivered interpersonally, at scale, in urban poor settings in LMICs. This scope is already described in the Introduction (lines beginning: “This scoping review aims to enhance understanding of how infectious disease interventions can be adapted to urban poor settings…”) and detailed in the Eligibility criteria and Table 1 – Type of intervention.

The scope of interventions we include is consistent with that used in our earlier Epling et al. review, which examined delivery to racially, ethnically, and religiously marginalized populations.

2. Could you also specify, at least throughout the text (if not here as well), whether the interventions you wish to assess are mass/population-based public health interventions as it seems, in opposition to individual-level interventions? Based on Table 1, it seems you also wish to assess the outcomes of those interventions. However, this information does not appear in this title and in the introduction as expected.

Our focus is on mass population-level interventions – that include interpersonal delivery. We have revised lines 101 for better clarity to read: “Given our focus on the delivery context, we will concentrate on a subset of population-based interventions characterized by one-on-one interactions…” ) and reiterated in Table 1 – Inclusion criteria under “Type of intervention.”

As noted in Table 1 – Reported outcome and in the Objective section, we intend to extract information on outcomes reported (e.g., coverage, cost, prevalence, knowledge/attitudes). However, we will not be assessing interventions based on these results, as this falls outside the scope of a scoping review. A systematic review would be required for that level of synthesis, and at this stage we do not believe such evidence is available. This information will be used to report out on the types of outcomes measured – to inform the design of future studies.

3. As mentioned below, the term “poor” could be considered to be discriminative and I would remove it, especially since urban areas in LMICs most often have limited all-type resources. See BMJ Glob Health. 2022 Jun;7(6):e009704. doi: 10.1136/bmjgh-2022-009704.

We appreciate this important point and acknowledge the concern about terminology. To address it, we will replace “urban poor” with “low-income urban communities” throughout the manuscript. This phrasing avoids defining people by what they lack while remaining consistent with our scope. As described in our Methods (search strategy), our review includes related descriptors used by study authors, such as “slums,” “informal settlements,” and “street dwellers,” to ensure comprehensiveness.

Editors comment on abstract

1. Conform to PLOS One author guidelines when formatting the revised manuscript. Accordingly, consider removing the Ethics statement in the abstract as well as sub-sections. Briefly, the abstract should look like in this article: https://doi.org/10.1371/journal.pone.0328645.

We agree and will reformat the abstract to align with PLOS ONE guidelines. Specifically, we will remove the subheadings and the ethics statement, and restructure the abstract to match the journal’s style. Please see the revised abstract

Editor comments on Keywords

1. Why not just write «Urban» throughout the manuscript instead of «Urban poor»? Is it not possible to have a non-poor urban area in LMICs? See BMJ Glob Health. 2022 Jun;7(6):e009704. doi: 10.1136/bmjgh-2022-009704.

We appreciate this important point and acknowledge the concern about terminology. To address it, we will replace “urban poor” with “low-income urban communities” throughout the manuscript.

More specific comments from Editor

C1: Introduction: The introduction is currently unnecessarily very long. The greatest length of this manuscript should be reserved to the Materials and Methods section below.

We agree that the Introduction could be more concise and have revised it accordingly.

C2: Line 56-57: Could you complete this statement? Where do people moving to urban areas in LMICs come from? Or where did they come from? Rural areas of LMICs or from urban areas of high-income countries? What in their previous living areas drove them to urban areas of LMICs? Why did they not move to urban r rural areas of high-income countries?

We appreciate this thoughtful set of questions. The drivers of urban migration in LMICs vary widely by country and context. As this is the study protocol we have not yet extracted the information on setting. We therefore do not propose changes to the current text in the introduction, however we will ensure that our data extraction framework (see Box 1 and Data Extraction section) is designed to capture contextual details described in the included studies, including migration patterns and community-level characteristics. This approach will allow us to better reflect the nuances of where populations come from and why, based on empirical evidence.

C3: Line 62: I would complete this sentence with a statement like: «owing to limited hygiene and therefore highly prone to the development of infectious outbreaks.» See https://doi.org/10.1038/s41579-021-00639-z

Drawing on the cover letter, the authors would also like to complete this statement with another one showing data about how rapid urbanization, increase in population density and the lack of health care infrastructures and resources drive the widespread of infectious diseases (IDs) in urban areas in LMICs.

We appreciate the suggestion. However, our review is not limited to outbreaks but is designed to capture the full range of infectious disease interventions delivered interpersonally in urban poor LMIC settings as explained above. The current text already highlights how rapid urbanization, high population density, and weak infrastructure create environments where infectious diseases thrive (see Introduction, second paragraph). To keep the Introduction concise (in line with recommendations made earlier) we prefer not to expand further.

C4: Line 63: Is this an urban area of a LMIC? If need be to take such an example alongside the above-suggested revisions, I would take the example of a prototypic highly populated urban area of a LMIC with increased ID burden (e.g., an urban area of Nigeria).

We appreciate this suggestion. We have removed these lines completely to ensure our introduction is concise as earlier suggested.

C5: Line66-67: Alongside the comment I made in line 62, this sentence would better be moved when presenting data about how rapid urbanization, increase in population density and the lack of health care infrastructures and resources drive the widespread of infectious diseases (IDs) in urban areas in LMICs. Once again, I suggest revising and completing this sentence to focus on urban areas of LMICs and on consequences of this fact with regard to the spread and control of infectious diseases.

This article may be helpful in this regard; https://doi.org/10.1038/s41579-021-00639-z

We thank the editor for this suggestion. As noted in our response to C3, our review is not restricted to outbreaks but covers a broader set of infectious disease interventions in urban poor LMIC settings. The current placement of this sentence in the Introduction is intended to show how rapid urbanization and weak infrastructure create conditions where infectious diseases thrive, thereby motivating the need for adapted interventions. While we recognize the value of framing in terms of outbreak vulnerability, shifting the focus in this way would change the scope of the review. We therefore propose to retain the current structure, while ensuring that our extraction framework captures how interventions were adapted in response to the full range of challenges faced in urban poor settings.

C6: It would also be great to end this comment with a few details about why the

I think this comment was cut off and we are unsure how to reply.

C7: While commenting on how urbanization in LMICs drives the spread of ID outbreaks, it would be great to mention whether and eventually why the problematic of ID control in rural areas of LMICs would be less relevant than that of ID control in urban areas of LMICs.

We thank the editor for raising this point. Most infectious disease control programs operate in rural areas, usually within existing primary health care units. However, our focus on urban poor settings does not imply that rural infectious disease control is less relevant. Rather, the delivery of interventions differs substantially between urban and rural contexts. Urban areas pose distinct challenges—including density, heterogeneity, and mobility—that require different forms of adaptation than those typically needed in rural settings. To keep the review focused and feasible, we have chosen to examine urban poor contexts, while recognizing that rural delivery remains equally important but outside the scope of this protocol.

C8: Line 69: If the authors implement the revisions suggested above, they would have highlighted how rapid urbanization with increase in population density and lack of health care infrastructures and other resources in urban areas of LMICs drive and sustain the widespread of ID outbreaks in urban areas of LMICs at this level of the introduction. Accordingly, from line 69 to line 109, they would better focus on the current state of knowledge about interventions to control IDs in urban areas of LMICs in one paragraph, and on gaps in knowledge about interventions to control ID outbreaks in urban areas of LMICs in another paragraph. When presenting the current state of knowledge and gaps of knowledge, the authors would like to address issues such as whether the interventions are tailored to the context and culture of the researched populations and how much they contributed to the development and implementation of those interventions, including the role of community leaders and local researchers. While doing this, the authors

1. DOI: 10.1016/S1473-3099(24)00744-8

2. https://doi.org/10.1186/s13643-024-02651-9

These articles would greatly help the authors frame the needed text:

https://doi.org/10.1186/s13643-024-02651-9

https://doi.org/10.1136/bmjgh-2019-001855

Bayingana et al. Lancet Glob Health 2025; 13: e1627–35.

https://doi.org/10.1038/s41579-021-00639-z

https://doi.org/10.1371/journal.pone.0328645.

https://doi.org/10.1371/journal.pone.0317177

We thank the editor for this helpful suggestion. We agree that the Introduction could be strengthened by distinguishing more clearly between (1) what is already known about infectious disease interventions in urban LMIC contexts, and (2) the remaining gaps in knowledge. We have thus revised our third paragraph along those lines.

C9: Line 97: This is rather a scoping review protocol.

We agree with this clarification. The manuscript is indeed a protocol for a scoping review, not a completed review. This is already indicated in the title (“…a scoping review protocol”) and reiterated in the Protocol Design section (line 120 onwards).

• In the Abstract, Materials and Methods we wrote:

“This scoping review will follow Joanna Briggs Institute (JBI) guidelines and the PRISMA methodology adapted for scoping reviews (PRISMA-ScR).”

• In the Protocol Design section we wrote:

“…we will conduct a scoping review using the Joanna Briggs Institute (JBI) guidelines [24] and utilize the PRISMA extension for scoping reviews (PRISMA-ScR) guidelines for reporting [25].”

C10: Line 111-117 (on objectives): Consider revising this text using a simple language (simple sentences). It seems to me that you would like to assess whether interventions used to control the spread of ID (outbreaks) so far in urban areas of LMICs were contextually and culturally appropriate or relevant. If this is the case, it should be clearly stated. You should also consider the rationale for conducting any scoping review (map the body of evidence on a given subject) when framing the aim of this scoping review protocol (not yet the scoping review). See https://doi.org/10.1186/s12874-018-0611-x

https://doi.org/10.1371/journal.pone.0328645

https://doi.org/10.1371/journal.pone.0317177

We thank the reviewer for this observation. The objective of our review is not to assess the effectiveness of interventions in outbreaks, but rather to m

---

## [Decision Letter · Decision Letter 1]

16 Nov 2025

Exploring how infectious diseases control interventions are delivered in low-income urban communities in in low- and middle-income countries: a scoping review protocol

PONE-D-25-40772R1

Dear Dr. Mogaji,

We’re pleased to inform you that your manuscript has been judged scientifically suitable for publication and will be formally accepted for publication once it meets all outstanding technical requirements.

Kind regards,

Mickael Essouma, M. D.

Academic Editor

PLOS ONE

Additional Editor Comments (optional):

There are a few details that need to be addressed during the production stage: edits mentioned in the attached document PONE-D-25-40772_R1_reviewed by Mickael Essouma.pdf and that references are correctly placed in the text.

Reviewers' comments:

Reviewer's Responses to Questions

**Comments to the Author**

1. Does the manuscript provide a valid rationale for the proposed study, with clearly identified and justified research questions?

Reviewer #1: Yes

2. Is the protocol technically sound and planned in a manner that will lead to a meaningful outcome and allow testing the stated hypotheses?

Reviewer #1: Yes

3. Is the methodology feasible and described in sufficient detail to allow the work to be replicable?

Reviewer #1: Yes

4. Have the authors described where all data underlying the findings will be made available when the study is complete?

Reviewer #1: Yes

5. Is the manuscript presented in an intelligible fashion and written in standard English?

Reviewer #1: Yes

You may also provide optional suggestions and comments to authors that they might find helpful in planning their study.

Reviewer #1: I feel the authors have worked on their manuscript. They have addressed my comments well. The manuscript rea\ds well and is ready for publication.

**Do you want your identity to be public for this peer review?** For information about this choice, including consent withdrawal, please see our Privacy Policy

Reviewer #1: No

---

## [Editor Report · Acceptance letter]

PONE-D-25-40772R1

PLOS ONE

Dear Dr. Mogaji,

I'm pleased to inform you that your manuscript has been deemed suitable for publication in PLOS ONE. Congratulations! Your manuscript is now being handed over to our production team.

Kind regards,

on behalf of

Dr. Mickael Essouma

Academic Editor

PLOS ONE